# Multi-Inflammatory Syndrome in Children (MIS-C) in 2023: Is It Time to Forget about It?

**DOI:** 10.3390/children10060980

**Published:** 2023-05-31

**Authors:** Francesco La Torre, Andrea Taddio, Chiara Conti, Marco Cattalini

**Affiliations:** 1Pediatric Rheumatology Center, Department of Pediatrics, Giovanni XXIII Pediatric Hospital, University of Bari, 70121 Bari, Italy; 2Institute of Child and Maternal Health–IRCCS “Burlo Garofolo”, University of Trieste, 34127 Trieste, Italy; 3Pediatrics Clinic, Department of Experimental and Clinical Sciences, University of Brescia, 25121 Brescia, Italy

**Keywords:** MIS-C, SARS-CoV-2 vaccine, children, IVIG, corticosteroids, anakinra, COVID-19

## Abstract

Multisystem inflammatory syndrome in children (MIS-C) is defined as a clinically serious condition requiring hospitalization involving fever, multi-system organ dysfunction, and an increase in inflammatory biomarkers. The syndrome was originally described as a post-infectious complication of severe acute respiratory syndrome coronavirus 2 (SARS-CoV-2) infection, which usually causes COVID-19. During the COVID-19 pandemic, not only did the virus undergo mutations but vaccines against SARS-CoV-2 were also developed. Both these conditions led to a decrease in the incidence of MIS-C. This narrative review summarizes the recent updates for MIS-C, particularly regarding the change in incidence, the link between the SARS-CoV-2 vaccine and MIS-C, and new updates of MIS-C treatments.

## 1. Introduction

In late 2019, a novel virus causing severe pneumonia was detected in China. The virus belongs to the group of coronaviruses and was designated severe acute respiratory syndrome coronavirus 2 (SARS-CoV-2). In February 2020, the World Health Organization called the disease resulting from infection with SARS-CoV-2 coronavirus disease 2019 (COVID-19) [1]. The virus rapidly spread globally and developed into a pandemic that, to date, has caused more than 6.5 million deaths [2]. In Europe, the first isolated cases were reported in France, and afterwards, a larger outbreak was detected in Italy [3]. In late April 2020, soon after the first wave of SARS-CoV-2 infections in Italy, the Rheumatology Study Group of the Italian Society of Pediatrics sent an alert to its associates on an unusual peak in the number of children presenting with a severe systemic inflammatory disease resembling Kawasaki disease (FKD) but with some atypical features, such as older age at onset, high incidence of cardiogenic shock and myocarditis, and abdominal symptoms [4,5]. The first case reports were published soon after by Verdoni et al. [6]. In the following weeks, more reports emerged from all over the world of this hyperinflammatory syndrome possibly related to SARS-CoV-2, which is nowadays called multisystem inflammatory syndrome in children (MIS-C) or pediatric inflammatory multisystem syndrome temporally associated with SARS-CoV-2 (PIMS-TS) [7,8,9,10,11,12,13,14,15,16]. Different case definition criteria have been proposed. (Table 1) The most consistent signs and symptoms required to diagnose MIS-C are the presence of fever at least 24 h prior to diagnosis, multiorgan involvement (for example, cardiocirculatory, gastrointestinal, hematologic, neurologic, mucocutaneous, hepatic, respiratory, or renal symptoms), the elevation of inflammatory markers, evidence of a recent COVID-19 infection, and exclusion of other possible etiologies [16,17,18,19] (Table 2). Unfortunately, MIS-C criteria are not very specific and it has been suggested that only a small proportion of children with the most severe manifestations of acute COVID-19 may satisfy them. Finally, MIS-C has, although very rarely, also been described in adults [20,21,22,23,24,25]. This narrative review aims to analyze the new evidence relating to MIS-C, particularly regarding the change in its incidence, the link between MIS-C and the anti-SARS-CoV-2 vaccine, and its treatment. We conducted multiple PubMed-types of research using a variable association of the terms: “MIS-C, Multiinflammatory Syndrome, COVID-19, SARSCoV2, SARSCoV2 vaccine” [and] “children”, and “MIS-C” [and] “adults”.

## 2. Pathogenic Hypothesis for MIS-C: The Role of Superantigens and Antibody-Dependent Enhancement 

MIS-C arises 4 to 6 weeks after prior SARS-CoV-2 exposure, which usually occurs as an asymptomatic or mild infection. Most patients test negative in nasopharyngeal reverse transcriptase–polymerase chain reaction (RT-PCR) tests but positive for the serology of anti-SARS-CoV2 antibodies, which suggests that a post-infectious immune response may play a major role in the disease, although its exact mechanism is still unknown [5,7,26]. This immune response is characterized by the drastic production of proinflammatory cytokines, mainly TNF alpha, IL-1, and IL6, along with a decrease in lymphocytes, such as NK cells, CD4 T lymphocytes, and B lymphocytes [27]. The release of this “cytokine storm” contributes to a hyperinflammatory state that culminates in endothelial dysfunction and multiorgan damage [28]. The excellent response to immunomodulatory therapies further underlines the key role of immunological disruption with the production of pro-inflammatory cytokines in MIS-C manifestations. Indeed, besides IVIG and glucocorticoids, IL-1 receptor antagonists have been shown to reduce disease severity as well as mortality in severe COVID-19 infections and MIS-C [28,29,30,31,32,33]. A role for endothelial damage in MIS-C is furthermore suggested by the detection of Burr cells and schistocytes in some patients showing renal failure and thrombotic microangiopathy, in whom eculizumab has been reported to be highly effective [34,35,36].

Other evidence points towards an antibody-dependent enhancement (ADE) as the underlying mechanism, as in other coronavirus infections, such as SARS and MERS [37]. ADE develops due to preexisting non-neutralizing antibodies resulting from a previous vaccination or infection, which can facilitate virus dissemination in the host [34]. In fact, the concentration of IgG antibodies in SARS-CoV-1 infections seems to correlate with disease severity, since higher levels have been detected in critical cases; hence, some assume that ADE may increase tissue damage. ADE could also explain findings of more elevated anti-spike antibody titers in children developing MIS-C compared to those with SARS-CoV-2 acute infection alone [38,39]. Notably, even patients affected by severe Dengue fever show a similar clinical picture to some MIS-C cases. As with coronaviruses, critical Dengue patients with higher titers of virus-specific antibodies show worse clinical outcomes [32]. Karthik et al. discussed the development of vaccines for SARS-CoV-2 and provided some strategies to prevent ADE: molecules targeting partial subunits of SARS-CoV-2 spike protein and the glycosylation of some epitopes could elicit the production of high-affinity neutralizing antibodies and, therefore, lead to a better immune response [40]. Another ADE-related risk is linked to mastcell degranulation, which may be involved in cardiac damage through the binding of SARS-CoV-2 antibodies to Fc receptors. This model is also plausible in infants with MIS-C mediated by maternally transferred antibodies [41].

Another pathogenic hypothesis is the involvement of superantigen motifs in SARS-CoV-2, suggested by the clinical overlap between MIS-C and toxic shock syndrome (TSS), the trigger of which is to be found in bacterial superantigens. Rivas et al. actually described the discovery of a superantigen-like structure in SARS-CoV-2 spike 1 (S1) glycoprotein that is highly similar to Staphylococcal enterotoxin B, known for being the cause of TSS [42]. Higher concentrations of S1 glycoprotein seem to correlate with both COVID-19 and MIS-C severity, which is likely linked to greater TCR signaling and consequent T cell activation. Indeed, MIS-C children have been described as carrying both CD4+ and CD8+ memory T cells with the significant skewing of the V-beta repertoire and association with specific HLAs, all data pointing to a possible oligoclonal expansion in response to a superantigen motif [27]. Given that some S1 segments undergo mutations in selected viral variants, the superantigen model could also explain the variability in MIS-C incidence in Europe and North America compared to Asia.

Finally, an autopsy study with children affected by severe COVID-19 with or without MIS-C identified viral dissemination in several tissue types, suggesting a direct influence of organ invasion and replication on the clinical outcome [43].

## 3. How Common Is MIS-C and Is MIS-C Incidence Decreasing through Different Viral Strains?

MIS-C appears to be a relatively rare complication of COVID-19 in children, with an incidence that was initially estimated to be between 54 and 45 cases/100,000 SARS-CoV-2 infections in children < 15 years old [44,45]. Since the first descriptions, the incidence of MIS-C has been decreasing over time [15]. In fact, several studies from all over the world agree on the lower incidence of MIS-C during the *Delta* and *Omicron* waves compared to the *Wuhan* and *Alpha* variants [46,47,48,49]. Since many of those studies were conducted before the large vaccination campaigns for children, this could possibly be related to the change in the virus strains and points to the direct role of the virus strain in triggering MIS-C. Additionally, USA studies analyzing data from February 2020 to January 2022 suggested a decrease in MIS-C severity. Cardiovascular complications and clinical outcomes, including length of hospitalization, the need for ECMO, and death, have shown improvement over time [50,51]. On the other hand, other outcome measures have not improved, and this effect on outcomes may have been determined by the prompter diagnosis and treatment of MIS-C as our knowledge of the disease has improved.

## 4. MIS-C and SARS-CoV-2 Vaccines

A reduction in the number of MIS-C cases could be related not only to the emergence of new virus strains but also to the vast vaccination campaign. Updated data about the administration of the vaccine reveal that the European cumulative uptake of the primary vaccination course has reached 72.8% (range: 29.9–86.5%), with percentages of 53.6% (range: 9.2–71.9%) for the first booster dose and 6.1% for the second booster dose (range: <0.1–16.3%) [52]. Among the US population, CDC data report that 445,786,989 children aged 5–18 years have completed the primary series (72.9% of the pediatric population between 5 and 12 years and 77% of the adolescents between 12 and 18 years) and 78.807.664 children have received the booster dose (12.7% of the individuals aged between 5 and 12 years and 13.8% of the remaining pediatric population under 18 years) [53]. A case–control study from the US during the *Delta* wave reported vaccine effectiveness in preventing MIS-C after two doses of Pfizer-BioNTech as 91% (95% CI = 78–97%). All severe MIS-C patients requiring life support have been unvaccinated [54]. A Danish nationwide prospective cohort study also investigated the vaccine effectiveness against the development of MIS-C during the *Delta* wave, showing 94% efficacy [55]. Some MIS-C cases caused by the vaccine itself were also reported, although the direct link between the vaccination and MIS-C is far from conclusive in most cases [56,57,58,59]. Yousaf et al. retrieved data from patients who developed MIS-C at some time after vaccination and without proof of a precedent SARS-CoV-2 infection. Even assuming all these cases were secondary to SARS-CoV-2 vaccines, the rate would be 0.3 MIS-C cases per million children vaccinated, a rate that is far lower than the rate of MIS-C occurring after the infection, making it a rare adverse event and still supporting the protecting role of the vaccination against the development of MIS-C [60]. The protecting role of the vaccination also emerged indirectly from the efficacy of the maternal COVID-19 vaccine against the development of MIS-C in infants younger than 6 months. Chetna Mangat et al., in their narrative review, suggest that maternal vaccination may give passive immunity to infants <6 months of age and lower their risk of developing MIS-C [61]. Another clue to consider on the link between SARS-CoV-2 and vaccination pertains to the safety of vaccines in patients with a history of MIS-C [62]. Indeed, when the SARS-CoV-2 vaccine became available for children, there were some concerns that vaccination might cause a MIS-C relapse in patients recovering from the disease [62]. There are now sufficient data from multiple cohorts to affirm that the SARS-CoV-2 vaccine is safe in patients who have already had MIS-C [63,64,65]. The Centers for Disease Control and Prevention (CDC) and the European Academy of Allergy and Clinical Immunology (EAACI) recommend considering vaccination for patients who have recovered from MIS-C (with the restoration of cardiac function) and at least 90 days after the MIS-C diagnosis, although the final decision has to be taken on a case-by-case basis [66,67].

## 5. Therapy 

Given the clinical overlap between MIS-C and Kawasaki disease, patients with MIS-C have been treated since the beginning of the pandemic with a variable combination of intravenous immunoglobulins (IVIG) and glucocorticoids [5,6,7,26,68]. This has also been ratified by various scientific societies, which have published clinical recommendations [67,69,70]. Few trials have tried to establish the best treatment approach for patients with MIS-C, particularly with regard to the association between glucocorticoids and IVIG. Son et al. found that the initial treatment with IVIG plus glucocorticoids was associated with a lower risk of new or persistent cardiovascular dysfunction than IVIG alone, while McArdle et al. found no evidence that the recovery from MIS-C differed after primary treatment with IVIG alone, IVIG plus glucocorticoids, or glucocorticoids alone [68,71]. These discrepancies are most probably due to the retrospective nature of the studies and patient heterogeneity. Ouldali et al., with a robust statistical approach, found that the combination of IVIG and glucocorticoids was superior to IVIG alone in terms of cardiac outcome [72]. This finding was incorporated into the last version of the American College of Rheumatology (ACR) recommendations for the treatment of MIS-C [73]. In the same recommendations, the use of biologics is suggested in patients with refractory disease. Indeed, biologics—and IL-1-blocking agents in particular—have been used anecdotally in patients with MIS-C with good results [74,75,76,77]. Interestingly, there is now mounting evidence of the role of IL-1 in driving the inflammatory process and the presence of neutralizing anti-IL1RA antibodies (that lead to increased secretion of IL-1 by inflammatory cells) in patients with MIS-C [78]. All this evidence further suggests the use of Kineret for the treatment of refractory MIS-C. 

## 6. Final Considerations

Even though the number of MIS-C cases has been constantly decreasing, we believe that the timely recognition and prompt treatment of MIS-C are crucial for better patient outcomes; therefore, a high alert level should be maintained among pediatricians. Furthermore, MIS-C represents a very good model of a hyperinflammatory disease with a possible multifactorial origin. Now that we are not in an ”emergency modality“, its study could be very useful to better understand the link between viral infection, antibody response, genetic susceptibility, and inflammation. Furthermore, a better understanding of MIS-C could help in suggesting possible treatment targets that may be useful for other diseases, such as KD.

## Figures and Tables

**Table 1 children-10-00980-t001:** A comparison of the MIS-C criteria proposed by the World Health Organization (WHO), the US Centers for Disease Control and Prevention (CDC), and the UK Royal College of Pediatrics and Child Health (RCPCH).

	MIS-C Criteria Proposed by World Health Organization (WHO)	MIS-C Criteria Proposed by Centers for Disease Control and Prevention (CDC)	MIS-C Criteria Proposed by Royal College of Pediatrics and Child Health (RCPCH)
Age	0–19 years	<21 years	Pediatric patient (age is not specified)
Fever	>3 days	Fever > 38 °C for over 24 h	Persistent fever > 38.5 °C
AND	I	involvement of >two organ systems	
Clinical features	(1) Rash or bilateral non-purulent conjunctivitis or mucocutaneous inflammation (2) Hypotension or shock (3) Myocardial dysfunction, pericarditis, valvulitis, or coronary abnormalities (including imaging or laboratory findings)	Evidence of clinically severe illness requiring hospitalization with multisystem organ involvement (cardiac, renal, respiratory, gastrointestinal, dermatologic, hematologic, or neurological)	Evidence of single- or multiorgan dysfunction (shock, cardiac, renal, respiratory, gastrointestinal, or neurological disorder)
Markers of inflammation	Elevated indexes, such as ESR, CRP, or procalcitonin	One or more of the following: elevated CRP, ESR, fibrinogen, procalcitonin, D-dimer, ferritin, LDH, or IL6; elevated neutrophils; reduced lymphocytes; low albumin	Neutrophilia, elevated CRP and lymphopenia, abnormal fibrinogen, elevated D-dimers or ferritin, hypoalbuminemia
Absence of other etiologies	No other obvious microbial cause of inflammation, including bacterial sepsis and staphylococcal or streptococcal shock syndromes	No alternative plausible diagnosis	Exclusion of any other microbial cause, including bacterial sepsis, staphylococcal or streptococcal shock syndromes, and infections associated with myocarditis, such as enterovirus
Evidence of COVID-19	COVID-19 (RT-PCR, antigen test, or serology positivity) or likely exposure to COVID-19 patients	Positive for current or recent SARS-CoV-2 infection (RT-PCR, antigen test, or serology positive) or exposure to suspected or confirmed COVID-19 case within the 4 weeks prior to the onset of symptoms	SARS-CoV-2 PCR testing may be positive or negative

**Table 2 children-10-00980-t002:** A birds-eye view of the common features of MIS-C compared with those of Kawasaki disease.

	Common Features of MIS-C	Common Features of Kawasaki Disease
Age	6–11 years	6 months to 5 years
Sex	No predominance	Male
Ethnicity	Higher incidence in African and Hispanic children	Higher incidence in East Asian countries
Trigger	Occurs 4–6 weeks after prior SARS-CoV-2 infection SARS-CoV-2 IgG antibodies are detected	Unknown, possibly preceded by viral or bacterial infection
Immunological characteristics	Enhancement of IL-1β pathway Frequent lymphopenia >50% of patients with MIS-C have a MAS-like cytokine phenotype	Enhancement of IL-1β pathway Lymphopenia is a rare event Lower incidence of MAS
Clinical features	Fever, rash, cervical lymphadenopathy, neurological symptoms, extremity changes High incidence of gastrointestinal symptoms, myocarditis and shock, and coagulopathy	Fever, rash, cervical lymphadenopathy, neurological symptoms, extremity changes High incidence of conjunctival injection and oral mucous membrane changes, common development of coronary artery aneurysms
Heart involvement	Most commonly myocarditis but also coronary artery aneurisms and pericarditis	Most commonly coronary artery aneurisms

## Data Availability

Not applicable.

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
