# Peer review of "Multi-Inflammatory Syndrome in Children (MIS-C) in 2023: Is It Time to Forget about It?"

_children, 2023, doi:10.3390/children10060980_

Round 1

Reviewer 1 Report

Please adapt the PRISMA statement/checklist to your review.

The modifications can include search terms, study inclusion/exclusion criteria, and quality assessment of studies.

Please also adapt the Meta-analysis Of Observational Studies in Epidemiology (MOOSE) statement/checklist.

Also practical is the Cochrane Handbook for Systematic Reviews of Interventions.

Plus

Rewrite the abstract, present one confusing and needs to mention COVID-19.

Please review and comment:

Godfred Cato S. MMWR 2020; 69:1074.

Ciochetto Z. BMC Infect Dis 2021; 21:1228.

Lopez Leon S. Sci Rep 2022; 12:9950.

Author Response

  1. Dear Reviewer, thanks for taking the time to review our manuscript. Although your suggestions have been taken into the utmost consideration, please consider that this is not a systematic review but a narrative review. For these reasons, we are not sure the PRISMA (and all other methodological modifications suggested) will apply here. We hope you will agree with us.
  2. We appreciate your suggestions on the abstract and modified it.
  3. We added the first two references suggested and commented in the text, we also added more references than the one suggested for the occurrence of MIS in adults, thanks a lot for pointing that out. As far as Lopez Leon et al. is concerned, this is a very nice review on long-COVID, a topic not discussed in our review, which is focused on MIS-C. I hope you will agree with our decision not to add this reference.

Reviewer 2 Report

Nice review of MIS-C and its changing incidence.

Author Response

Thank you very much for taking the time to review our work and for your nice comment

Reviewer 3 Report

Thanks for the opportunity to review this paper. This review was described “Multi-inflammatory Syndrome in Children (MIS-C) in 2023: is it time to forget about it?”.  This is clearly well-written and documented review. I devoured the manuscript.  I congratulate to author.  A review that summarizes the subject mentioned very perfectly.

-       Introduction section should be elaborated.  The objective of this study should be clearly explained. 

-       The studying method should be briefly summarized.

-       References section should be updated. References section should be written according to the rules. Authors should be cited this reference. 

§  Haslak F et al. A cursed goodbye kiss from severe acute respiratory syndrome-coronavirus-2 to its pediatric hosts: multisystem inflammatory syndrome in children. Curr Opin Rheumatol. 2023 Jan 1;35(1):6-16.

§  Haslak F et al. A recently explored aspect of the iceberg named COVID-19: multisystem inflammatory syndrome in children (MIS-C). Turk Arch Pediatr. 2021 Jan 1;56(1):3-9.

-       There are minimal spelling mistakes in the manuscript and reference section. These must be corrected.

-       All abbreviations should be summarized in this manuscript.  

-       Table headings should be more informative.

Author Response

  • We would like to thank the reviewer for taking the time to read our manuscript and for the very generous comments

  • Introduction section should be elaborated.  The objective of this study should be clearly explained. Thanks for the suggestion. We edited the intro to clearly express our objective

  • The studying method should be briefly summarized. Thanks for the comment. A brief description of the terms used for PubMed research was added.

  • References section should be updated. The references section should be written according to the rules. Authors should be cited this reference. Done, thanks for the suggestion

Reviewer 4 Report

This review summarizes the recent updates on MIS-C, with particular regard to its changed incidence, the link with the SARSCoV2 vaccine, and its therapy. Even though the number of MIS-C cases has been constantly reducing, the authors emphasized that the timely recognition and prompt treatment of MIS-C are crucial for a better patient outcome, and therefore a high alert level should be maintained.

It is very interesting review and is worthwhile for publish in Cardiology in children.

To further strengthen the manuscript, following points should be clarified and addressed.

There is no explanation in table2 key points of MIS-C and KD. That should be mentioned precisely, especially the difference of cardiac involvement between MIS-C and KD. Are there any pathological findings in coronary arterial wall in MIS-C?

Author Response

  • First of all we would like to thank the reviewer for taking the time to read our manuscript and for the nice general comment

  • There is no explanation in table2 key points of MIS-C and KD. That should be mentioned precisely, especially the difference of cardiac involvement between MIS-C and KD. Are there any pathological findings in coronary arterial wall in MIS-C?

    Thanks to the comment that allowed us to expand our table, emphasizing that, while in MIS-C the most common finding is myocarditis, coronary artery aneurisms are the most common manifestations of KD. As we know, there is only one report on the pathological findings in patients with MIS-C (EClinicalMedicine (2021) 35:100850 – already cited in the text). Among the patients reported only one had CAAs but, unfortunately, there is no report on the pathological findings on coronary tissues. Thanks again for pointing out this very relevant issue

Round 2

Reviewer 3 Report

All corrections and revisions are appropriate for me.